# Prospects of Developing Novel Genetic Resources by Chemical and Physical Mutagenesis to Enlarge the Genetic Window in Bread Wheat Varieties

**Momina Hussain** [1,2,3], **Maryyam Gul** [1,2], **Roop Kamal** [4], **Muhammad Atif Iqbal** [1,3], **Sana Zulfiqar** [1,2], **Ammad Abbas** [1,5], **Marion S. Röder** [4], **Quddoos H. Muqaddasi** [4,6,*] and **Mehboob-ur-Rahman** [1,2,*]

1  Plant Genomics and Molecular Breeding Laboratory, National Institute for Biotechnology and Genetic Engineering (NIBGE), P.O. Box 577, Jhang Road Faisalabad, Faisalabad 38000, Pakistan; momina_hussain5@yahoo.com (M.H.); soft.listener@hotmail.com (M.G.); atif1642@gmail.com (M.A.I.); sana_zulfiqar19@yahoo.com (S.Z.); Ammad.Abbas@ed.ac.uk (A.A.)
2  Department of Biotechnology, Pakistan Institute of Engineering and Applied Sciences (PIEAS), Nilore, Islamabad 45650, Pakistan
3  Department of Biotechnology, Faculty of Life Sciences, University of Okara, Pakistan
4  Leibniz Institute of Plant Genetics and Crop Plant Research (IPK), Corrensstraße 3, D-06466 Stadt Seeland, OT Gatersleben, Germany; kamal@ipk-gatersleben.de (R.K.); roder@ipk-gatersleben.de (M.S.R.)
5  Institute for Molecular Plant Sciences School of Biological Sciences, University of Edinburgh, Edinburgh EH9 3BF, UK
6  Present address: European Wheat Breeding Center, BASF Agricultural Solutions GmbH, Am Schwabeplan 8, D-06466 Stadt Seeland, OT Gatersleben, Germany
*  Correspondence: muqaddasi@ipk-gatersleben.de (Q.H.M.); mehboob_pbd@yahoo.com or mehboob@nibge.org (M.-u.-R.)

**Abstract:** Sustainable production and improved genetic gains can be achieved by broadening the genetic window of elite wheat germplasm. Here, we induced mutations in two spring wheat varieties, viz., NN-Gandum-1 (NN-1) and Punjab-11 (Pb-11), by exposing their seeds to ethyl methane sulfonate (EMS) and γ-rays, respectively. We characterized >3500 lines of each NN-1 and Pb-11 derived population in three consecutive generations, viz., $M_5$, $M_6$, and $M_7$, for important traits, e.g., plant height, heading date, spike morphology and rust resistance. We observed significant genetic variation and correlations in both populations for all investigated traits. We observed differences in terms of number of mutants between NN-1 (22.76%) and Pb-11 (26.18%) which could be ascribed to the genotype-by-mutagen interaction. High broad-sense heritability ($H^2$) estimates, that are vital for higher genetic gains, were observed for all of the investigated traits in both populations ($H^2$ = 0.69–0.91 in NN-1 and 0.84–0.98 in Pb-11). Particularly, to breed for rust resistance, we selected a subset ($n$ = 239) of $M_7$ lines that also showed phenotypic variation for other traits. Our studies (1) show the relevance to artificial mutagenesis to create genetic variation in elite germplasm for their immediate use in current breeding programs, and (2) provide material for downstream identification of genes associated with traits of high agronomic importance.

**Keywords:** *Triticum aestivum* L.; EMS; gamma radiation; z-score; mutagen-effectiveness; heritability; genetic resources

## 1. Introduction

Bread wheat (*Triticum aestivum* L.), a major source of daily caloric intake, is grown in more than 60 countries on 221.86 million ha with an annual production of 775.8 million metric tons (https://apps.fas.usda.gov/psdonline/circulars/production.pdf accessed on 1 June 2021) during 2020-21. The most cultivated wheat is allohexaploid ($2n$ = 6x, AABBDD) that harbors a large genome of the size ~17 Gb [1], and was originated recently through polyploidization and hybridization of *Triticum turgidum* (tetraploid, AABB genome) and *Aegilops tauschii* (diploid, DD genome) [2].

Significant gains in grain yield were achieved by introducing dwarfing (*Rht*) genes into wheat varieties using conventional breeding approaches that laid a firm foundation for the onset of the Green Revolution [3]. Improved germplasm harboring the *Rht* genes was introduced in several countries including Pakistan and India, and was extensively used for developing new wheat varieties, which consequently led to a significant production increase worldwide. Nevertheless, there are still many biotic and abiotic stress-associated challenges which result in decreased wheat production worldwide [4–6].

Among biotic stresses, rust diseases (e.g., yellow, leaf, and stem rusts) are the most detrimental to wheat production. Leaf rust, caused by *Puccinia triticina* f. sp. *tritici*, is distributed widely throughout the world [7] and reduces yield up to 70% by negatively impacting on the number of spikes per plant and grain weight [8–10]. Similarly, yellow rust, caused by *Puccinia triiformis* f. sp. *tritici* (*Pst*), may reduce yield up to 60% [11]. Stem rust, caused by *Puccinia graminis* f. sp. *triticiis*, is a disease of relatively warmer climates, which is prevalent in many wheat growing areas and, therefore, countries such as Pakistan and India, are at a high risk. Yield losses due to stem rust range usually from 10 to 45% but can be catastrophic under extreme conditions [12]. Additionally, evolution of new virulent strains may overcome the resistance of the cultivated varieties; this further complicates the situation [13,14]. Abiotic stresses, e.g., drought and salinity, on the other hand, cause wheat reduction by 50–90% and 10–90%, respectively [15–17]. These stresses impede plant growth and result in premature leaf senescence, reduced tillering and, consequently, reduced grain yield [5,18,19].

The challenges associated with both biotic and abiotic stresses can be addressed by developing diverse genetic resources that are required not only for developing resilient wheat cultivars using conventional breeding approaches, but also can be used for understanding the genetics of yield and yield component traits using modern genomic technologies [20,21]. The extent of genetic variations was found to be low among cultivated wheat varieties largely due to the development of varieties by repeatedly intercrossing of highly adaptive germplasm [4,22].

Mutagenesis is a widely used, effective, and economical means to induce genetic variations [23,24]. Chemical mutagens have been widely used for undertaking forward and reverse genetic studies for understanding the function of genes. The most commonly used chemical mutagens are ethyl methane sulphonate (EMS) and dimethyl sulfoxide (DMS), which generate random point mutations in the entire genome [25]. Physical mutagens, e.g., $\gamma$-rays, have also been used to create variations in various plant species. Optimal dose of the radiation is the key to achieving the maximum mutagenesis rate with an acceptable fertility rate of $M_1$ plants. However, low to moderate doses of the mutagen ensures the maximum frequency of beneficial mutants [26]. A high dose of radiations, on the other hand, may depress the viability of plants due to a greater ratio of deletions and aberrations in genes. Large deletions were, nevertheless, not carried to the $M_2$ generation of Arabidopsis, possibly due to loss of many genes controlling the development of the gamete and seed [27]. However, tolerance to large number of mutations, including deletions, was reported high in polyploids [28]. The definition of an effective mutagen relates to the production of relatively less biological damage (seedling injury, sterility, etc.) than the number of induced mutations in the genome. Thus, the choice of mutagen is important for inducing a high frequency of mutations.

In the present study, three generations (i.e., $M_5$, $M_6$ and $M_7$) of two mutant populations were developed by exposing the kernels of NN-Gandum-1(NN-1) and Punjab-11 (Pb-11) with EMS and $\gamma$-rays, respectively. We investigated the mutagenesis efficiency of chemical (EMS) and physical mutagens ($\gamma$-rays), genetic variability, and correlation among the investigated traits and generations. Lastly, we compared the effectiveness of EMS and $\gamma$-rays in the $M_7$ generations. The newly developed mutant germplasm is expected to (1) develop resilient wheat varieties, and (2) serve as a potential source to detect new alleles which can be used to provide insights into several mechanisms controlling traits of high agronomic and disease importance.

## 2. Materials and Methods

### 2.1. Genetic Material

Two stable bread wheat varieties NN-Gandum-1 (NN-1, approved for cultivation in 2016) and Punjab-11 (Pb-11, approved for cultivation in 2011) were exposed to chemical and physical mutagens. NN-1 is a high-yielding variety, but moderately-resistant moderately-susceptible (MRMS) to leaf rust (LR) and yellow rust (YR). It was developed by the Plant Genomics and Molecular Breeding Laboratory (NIBGE) Faisalabad Pakistan. Pb-11 is resistant to lodging and tolerant to leaf rust and was developed by the Wheat Research Institute (AARI), Faisalabad, Pakistan. Pb-11 is also susceptible to yellow rust and aphids, and is not suitable for late sowing, especially after cotton and rice.

### 2.2. Development of Mutant Populations

Breeder seeds of NN-1 were exposed to an optimized concentration of 0.8% (*v/v*) of ethyl methane sulphonate (EMS) for 2 h at 35 °C. For mutagenesis with γ-rays, the breeder seeds of Pb-11 were exposed to an optimum dose of 250k rad, Cs 137 for 25 min at room temperature (unpublished data, Figure 1). Seeds were sterilized with 5% sodium hypochlorite and 70% ethanol followed by three washings to remove their residues.

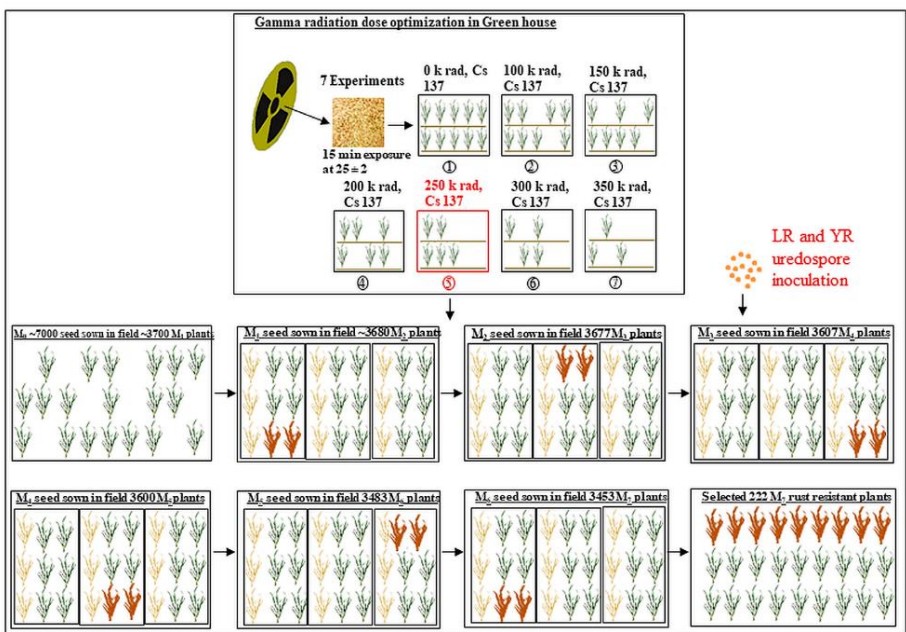

**Figure 1.** Optimization of gamma radiation dose and development of Punjab-11 population. Experiment 5 was selected as an optimized dose on the basis of germination rate (45–55%). Yellow colored plants were tagged from each row to record morphological data and collection of spikes. Red colored plants were wild type plants (nontreated), whereas green colored plants were the mutant population (treated with 250 k rad, Cs 137). LR and YR represent leaf and yellow rusts. Each generation (from M2 to M7) was inoculated with rust to ensure maximum disease attack.

A total of ~7500 seeds of NN-1 and ~7000 seeds of Pb-11 were exposed to the optimized concentration of EMS and γ-rays, respectively. The $M_0$ seeds of both the parental genotypes were manually sown in the experimental fields of NIBGE (Faisalabad, Pakistan) in 2012 by keeping 10 cm hill-to-hill distance and 30 cm between the rows. Standard agronomic practices were applied from sowing up to harvesting. In total, ~4000 and 3700 $M_1$ plants of NN-1 and Pb-11 were germinated, respectively. At maturity, one main spike of each plant was bagged and harvested separately. The harvested spikes were threshed followed by sowing of $M_2$ rows of each $M_1$ spike in 2013. The next year, $M_3$ populations were sown followed by harvesting a single spike from the main tiller. The $M_4$ seed of these mutant populations were planted in June 2014 at Naran and Kaghan (off-season nursery)

for advancing generation. A single plant with 3–5 tillers from each row was harvested separately. In 2014–2015, M$_4$ seeds were sown and harvested. In 2015–2018, the M$_5$ populations of NN-1 (3634 lines) and Pb-11 (3600 lines), M6 populations of NN-1 (3533 lines) and Pb-11 (3483 lines) and M$_7$ populations of NN-1 (3502 lines) and Pb-11 (3453 lines) were sown in NIBGE fields in a completely randomized design. The first and every 10th row was planted with the universal susceptible variety "Morocco" to ensure the provision of maximum inoculum.

### 2.3. Rust Inoculation

For producing the rust inoculum, uredospore (5 g) of leaf rust were taken in small petri plates from the infected leaves of the universal susceptible check "Morocco". These spores were thoroughly mixed in a solution (1 L distilled water with ~1 ppm Tween20). Concentration of the uredospore was estimated using a Hemocytometer. Final concentration of the solution was adjusted to $5 \times 104$ by adding distilled water. The solution was sprayed using a sprayer fixed with a fine nozzle on each plant at 16:00 h followed by the showering of water twice a day for two weeks. The same procedure was adopted for inoculating yellow rust spores. All generations (M$_2$ to M$_7$) were inoculated with the uredospore of yellow and leaf rust.

### 2.4. Collection of Data from Mutant Generations

Data on response to leaf and yellow rusts (LR and YR), plant height (PH; cm), tillers per plant (TPP), days to heading (DTH) after germination, and spike length (SL; cm) were recorded. Data on disease response were collected at maturity by following a 0–9 rating system [29] of all the mutant generations. Heading date (days after germination) was observed when 50% of the total spikes in each row emerged. Similarly, plant height was measured from the base to the top of a spike of each plant. Spike length were taken by measuring the length of spike from bottom to tip without awns.

### 2.5. Selection Strategy

Two selection strategies were used to select mutant lines from the individual populations. First, lines were selected on the basis of their reaction to LR and YR. This process resulted in selection of 239 lines from both populations (17 LR and YR resistant lines of NN-1 and 222 LR and YR resistant lines of Pb-11). Secondly, the experimental field was divided into small blocks (containing 100 lines in each) for minimizing the impact of micro-environment. Then the means and standard deviations were estimated for each of the subplots. After this, Z scores were computed for each of the 239 selected mutant lines as:

$$Z = (x - u)/sd \tag{1}$$

where *x*, *u* and *sd* denote the disease score, mean value and the standard deviation, respectively. The lines exhibiting the maximum Z score values for DTH, SL, PH and TPP were selected. This process resulted in the selection of lines with traits such as DTH (9 NN-1, 14 Pb-11), SL (12 NN-1, 84 Pb-11), PH (12 NN-1, 137 Pb-11) and TPP (6 NN-1, 52 Pb-11). In 2017–2018, the selected mutant lines were grown along with the wild type on 15 December 2017 in a randomized complete block design with three replications at two locations viz., NIAB and NIBGE, in Faisalabad. The plot size was 64 ft$^2$ (8 × 8 ft). The rust inoculum collected from the field was sprayed two times in January to March. Standard agronomic practices were applied from sowing to harvesting. At maturity, data regarding disease response and severity of DTH, PH, SL, LR, YR, and TPP were collected.

### 2.6. Mutagen Effectiveness

For studying the effectiveness of both mutagens (EMS and γ radiation), traits of each M$_7$ line of both populations were compared with the corresponding traits of the respective non-mutagenized (wild type) parent. For example, the EMS-treated population of NN-1 was compared with the non-treated NN-1, while the γ radiation population was compared

with the non-treated Pb-11 population. We compared six traits including DTH, PH, SL, TPP, LR and YR. The percentage of the number of mutants for a particular trait of both populations was calculated as:

$$\text{Mutant \%} = \frac{\text{Total number of mutants for a traits}}{\text{Total size of mutant population}} \times 100 \tag{2}$$

Gain over of each mutagen for a particular trait was calculated by subtracting the mutant % of NN-1 for each trait from the mutant % of the corresponding trait of the Pb-11 mutant population. Then we summed up the gain for each of the traits for both populations. The bigger value (of g-treated population) was subtracted from the smaller value of EMS-treated NN-1 population.

### 2.7. ANOVA, Heritability, and Correlation Analyses

We performed ANOVA to calculate the individual variance components of the genotype, environment, and the residuals. Pearson's product moment correlation was also performed for all investigated traits by using Statistix 8.1 and SPSS 16 software. Similarly, differences among the means of DTH, SL, PH and TPP were calculated using least significant difference. Plot-based broad-sense heritability of the investigated traits was calculated as

$$H^2 = \frac{\sigma_G^2}{\sigma_G^2 + \sigma_e^2} \tag{3}$$

where $\sigma_G^2$ and $\sigma_e^2$ represent the genotypic and the residual variance, respectively.

### 3. Results

In the present studies, significant variations in DTH, SL, PH and TPP were observed in both mutagenized populations. These differences were significantly higher than variations that could occur by chance. For example, in 2015–2016, $M_5$ lines derived from NN-1 showed variations in DTH (−4.43 to + 4.57 standard deviation (SD)), SL (−6.07 to +2.33 SD), PH (−16.04 to +10.96 SD) and TPP (−0.94 to +1.06 SD) (Table 1). Likewise, in 2015–2016, Pb-11-derived mutant lines ($M_5$) showed high variations for DTH (−1.97 to + 4.03 SD), SL (−5.27 to +2.01 SD), PH (−8.62 to + 4.49 SD) and TPP (−0.92 to +1.08 SD) (Table 1). Similar fluctuations in variations for each trait in both mutagenized populations were also recorded in 2016–2017 and 2017–2018 (Table 1). Relatively higher number of plants conferring resistance to LR and YR were observed in all mutant populations of Pb-11 than those of the mutant populations derived from NN-1. For instance, 17 (0.49%) and 222 (6.43%) resistant mutants of NN-1 and Pb-11 were observed, respectively. A large number of disease-susceptible mutants were found in the mutant population derived from NN-1.

**Table 1.** Generation-wise mean, range, and standard deviation (SD) for the investigated traits for NN-1 and Punjab-11.

| Traits | Population | Mean (SD) of $M_5$ | Mean (SD) of $M_6$ | Mean (SD) of $M_7$ | SD |
|---|---|---|---|---|---|
| NN-1 | | | | | |
| | 10 WT lines | 3634 | 3533 | 3502 | |
| DTH (days) | 89 | 92.57 (1.68) | 92.82 (1.62) | 92.4 (1.96) | −4.43 to + 4.57 |
| SL (cm) | 9.25 | 9.58 (0.68) | 9.67 (0.66) | 9.36 (0.97) | −6.07 to +2.33 |
| PH (cm) | 89.26 | 89.32 (4.08) | 89.52 (5.06) | 86.27 (5.58) | −16.04 to +10.96 |
| TPP | 5 | 5.06 (0.62) | 5.13 (0.57) | 5.07 (0.59) | −0.94 to +1.06 |
| Punjab-11 | | | | | |
| | 10 WT lines | 3600 | 3483 | 3453 | |
| DTH (days) | 92 | 93.03 (1.30) | 93.12 (1.31) | 93.19 (1.33) | −1.97 to + 4.03 |
| SL (cm) | 8.86 | 9.29 (1.05) | 9.3 (1.05) | 9.35 (1.05) | −5.27 to +2.01 |
| PH (cm) | 87.65 | 88.03 (3.13) | 88.15 (3.15) | 88.21 (3.13) | −8.62 to + 4.49 |
| TPP | 4 | 5.08 (0.63) | 5.1 (0.65) | 5.13 (0.62) | −0.92 to +1.08 |

DTH = days to heading; SL = spike length; PH = plant height; TPP = number of tillers per plant.

### 3.1. Prospects of NN-Gandum-1 Derived Population

A total of 3634 $M_5$, 3533 $M_6$, and 3502 $M_7$ mutant lines derived from NN-1 were observed for morphological traits. In total 659 $M_5$ (18.14%), 597 $M_6$ (16.9%) and 590 $M_7$ (16.85%) mutant lines showed substantial variations (85–97 days) in days-to-heading (DTH) at both the extremes (early and late) than those of the wild type (Table S1). Similarly, 929 $M_5$ (25.56%), 841 $M_6$ (23.8%) and 813 $M_7$ (23.21%) mutant lines expressed variations in plant height (PH) which ranged from 81.56 to 104.34 cm versus 89.26 cm for the wild type. Similarly, variations in spike length (SL) ranged from 7.85 to 14.65 cm. For example, the SL of 396 $M_5$ (10.9%), 352 $M_6$ (9.96%) and 335 $M_7$ (9.57%) mutant lines was shorter than the wild type (9.25 cm), while SL of 110 $M_5$ (3.03%), 106 $M_6$ (3%) and 95 $M_7$ (2.71%) mutant lines was larger than the wild type (Table S1). Regarding tillers per plant (TPP), 512 $M_5$ (14.09%), 505 $M_6$ (14.29%) and 498 $M_7$ (14.22%) mutant lines produced four TPP in comparison to five TPP in the wild type. On the other hand, 726 $M_5$ (19.98%), 712 $M_6$ (20.15%) and 692 $M_7$ (19.76%) mutant lines exhibited six TPP (Table S1). Since the data regarding response to leaf and yellow rust were also collected, in total, 33 $M_5$ (0.91%), 28 $M_6$ (0.79%) and 17 $M_7$ (0.49%) mutant lines showed high resistance to LR, whereas 1383 $M_5$ (38.06%), 1356 $M_6$ (38.38%) and 1349 $M_7$ (38.52%) mutant lines showed response like wild type (moderately resistant to moderately susceptible). A total of 80 $M_5$ (2.20%), 76 $M_6$ (2.15%) and 75 $M_7$ (2.14%) mutant lines were found highly susceptible to LR (Table S1). Regarding YR, in total, 39 $M_5$ (1.07%), 25 $M_6$ (0.71%) and 17 $M_7$ (0.49%) mutant lines were found resistant to YR, while 1343 $M_5$ (36.96%), 1330 $M_6$ (37.65%) and 1324 $M_7$ (37.81%) mutant lines showed responses like the wild type (moderately resistant), while 74 $M_5$ (2.04%), 68 $M_6$ (1.92%) and 67 $M_7$ (1.91%) mutant lines showed high susceptibility to YR (Table S1).

### 3.2. Prospects of Punjab-11-Derived Population

In a mutant population derived by exposing the seed of Punjab-11 (Pb-11) with γ radiation, a total of 3600 $M_5$, 3483 $M_6$ and 3453 $M_7$ mutant lines were studied for changes in morphological traits. A total of 762 $M_5$ (21.16%), 688 $M_6$ (19.76%) and 662 $M_7$ (19.17%) mutant lines showed substantial variations (86–99 days) in DTH at both the extremes compared to the wild type (Table S1). Similarly, 1208 $M_5$ (33.55%), 1125 $M_6$ (32.3%) and 1102 $M_7$ (31.91%) mutant lines expressed PH ranging from 80.65 to 100.36 cm as compared to the wild type (87.65 cm). The SL of 629 $M_5$ (17.47%), 515 $M_6$ (14.79%) and 506 $M_7$ (14.65%) mutant lines was shorter than the wild type (8.86 cm), while the SL of 417 $M_5$ (11.58%), 407 $M_6$ (11.69%) and 406 $M_7$ (11.76%) mutant lines was larger than the wild type (8.86 cm, Table S1). In total, 594 $M_5$ (16.5%), 511 $M_6$ (14.67%) and 495 $M_7$ (14.34%) mutant lines consisted of three TPP versus four TPP for wild type), while 579 $M_5$ (16.08%), 553 $M_6$ (15.88%) and 552 $M_7$ (15.99%) mutant lines were comprised of five TPP (higher than the wild type). The remaining population (~67%) was similar to the wild type (four TPP, Table S1). In Pb-11, as for NN-1, the phenotypic notes were also taken for leaf and yellow rust incidence. In total, 341 $M_5$ (9.47%), 263 $M_6$ (6.78%) and 222 $M_7$ (6.43%) mutant lines showed high resistance to LR, whereas 1401 $M_5$ (38.92%), 1398 $M_6$ (40.14%) and 1393 $M_7$ (40.34%) mutant lines were found similar to the wild type (moderately resistant). However, none of the mutant lines were found highly susceptible in all of the three generations (Table S1). A total of 301 $M_5$ (8.36%), 231 $M_6$ (6.63%) and 222 $M_7$ (6.43%) mutant lines showed resistance to YR, whereas 1405 $M_5$ (39.03%), 1390 $M_6$ (39.91%) and 1385 $M_7$ (40.11%) exhibited disease reactions like the wild type (resistant to moderately resistant). None of the mutant lines were found susceptible to the disease in all three generations (Table S1).

### 3.3. Correlations among Generations

Correlation coefficients studies were performed on whole mutant population for all the traits except LR and YR. Correlation coefficients of the traits among the mutant generations of NN-1 ranged from 0.36 to 0.69, while they were 0.79 to 0.99 for Pb-11 (Table 2). Correlation coefficients among generations for all the traits were found to be

significant. In both the mutant populations, correlation values for different traits in $M_6$ versus $M_7$ were higher than that of the $M_5$ versus $M_6$ (Table 2).

**Table 2.** Correlation between generations among all four traits for NN-1 and Pujnab-11.

| Genotypes | Generation | DTH (days) | SL (cm) | PH (cm) | TPP |
|---|---|---|---|---|---|
| | $M_5$ versus $M_6$ | 0.56 * | 0.65 * | 0.55 * | 0.36 * |
| NN-1 | $M_5$ versus $M_7$ | 0.52 * | 0.68 * | 0.5 * | 0.48 * |
| | $M_6$ versus $M_7$ | 0.59 * | 0.69 * | 0.61 * | 0.53 * |
| | $M_5$ versus $M_6$ | 0.99 * | 0.92 * | 0.79 * | 0.96 * |
| Pb-11 | $M_5$ versus $M_7$ | 0.99 * | 0.91 * | 0.79 * | 0.97 * |
| | $M_6$ versus $M_7$ | 0.99 * | 0.93 * | 0.79 * | 0.97 * |

DTH = days to heading; SL = spike length; PH = plant height; TPP = number of tillers per plant; * = significance at the 0.05 probability (*p*) level.

In the present study, DTH, SL, PH and TPP showed significantly positive correlations among three generations of both mutant populations. DTH showed strong significant positive correlations with TPP and PH with SL (Figure 2). In total, 239 resistant mutant lines were selected to study Z-score analysis. The Z-score of 17 selected mutant lines of NN-1 for DTH, SL, PH and TPP ranged from −1.42 to +1.54, −1.14 to +2.98, and −0.62 to +1.60 and −0.12 to +1.58, respectively. (Table S2), while the Z score of 222 selected mutant lines of Pb-11 for DTH, SL, PH and TPP was in the range of −1.65 to +1.36, −1.97 to +2.12, −1.46 to +2.38 and −1.82 to +1.40, respectively (Table S3).

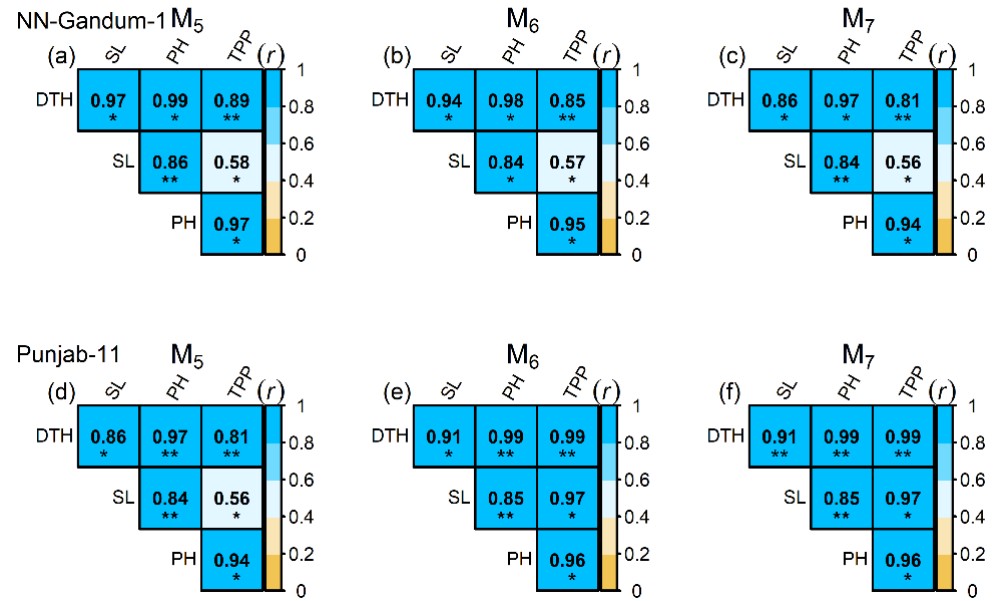

**Figure 2.** Pearson's product-moment correlation between traits between the years 2015–2018 for NN-1 (**a**–**c**) and Punjab-11 (**d**–**f**). DTH = days to heading; SL = spike length; PH = plant height; TPP = tillers per plant; * and ** = significance of the correlation at 0.05 and 0.01 probability (*p*) level, respectively.

*3.4. Comparison of the Selected Mutant Lines with the Wild-Type Parents*

3.4.1. Days to Heading

In total, nine $M_7$ and 14 $M_7$ lines were selected from 17 $M_7$ and 222 $M_7$ mutant lines of NN-1 and Pb-11 for DTH, respectively. The mean of selected lines of NN-1 for DTH was 93.05 days versus 89 days for the wild type (Table 3). However, only two lines, i.e., "#701" and "#1621" started flowering at 85 and 88 days after germination, respectively (Table S2). For Pb-11, the mean of the selected mutant lines for DTH was 92.06 days versus 92 days for the wild type (Table 3). A total of 12 lines (P-684, P-3339, P-646, P-539, P-572, P-21, P-685,

P-3078, P-3080, P-3081, P-3243 and P-3326) headed two days earlier than the wild type (Table S3).

**Table 3.** Z-score values of selected mutant plants from the M7 generation of NN-1 and Pb-11 at NIBGE and NIAB.

| Population | Parameter | DTH (Days) | SL (cm) | PH (cm) | TPP |
|---|---|---|---|---|---|
| NN-1 | Wild type | 89 | 9.25 | 89.26 | 5 |
| | Selected lines | 93.05 | 9.81 | 90.24 | 5.23 |
| | SD | 1.8 | 1.2 | 5.3 | 0.7 |
| | Z score range | −1.42 to +1.54 | −1.14 to +2.98 | −1.71 to +1.60 | −1.18 to +1.58 |
| | Lines with highest Z score | 9 | 12 | 12 | 6 |
| Pb-11 | Wild type | 92 | 8.86 | 87.65 | 4 |
| | Selected lines | 92.06 | 9.09 | 88.37 | 4.68 |
| | SD | 1.41 | 1.13 | 3 | 0.82 |
| | Z score range | −1.65 to +1.36 | −1.97 to +2.12 | −1.46 to +2.38 | −1.82 to +1.40 |
| | Lines with highest Z score | 14 | 81 | 137 | 52 |

DTH = days to heading; SL = spike length; PH = plant height; TPP = tillers per plant; SD = standard deviation.

### 3.4.2. Spike Length

In total, 12 $M_7$ and 81 $M_7$ lines were selected from 17 $M_7$ and 222 $M_7$ mutant lines of NN-1 and Pb-11, respectively. For NN-1, the mean SL of the selected lines was 9.81 cm versus 9.25 cm for the wild type (Table 3). Among these, the SL of three lines (i.e., 121, 1621, and 1929) were 11.45 cm, 11.45 cm and 12.25 cm versus 9.25 cm for the wild type, respectively (Table S2). For the mutant lines derived from Pb-11, the mean SL of the selected mutant lines was 9.09 cm while the wild type was 8.86 cm (Table 3). Mean SLs expressed by mutant lines P-684, P-685 and P-3077 were 11.25 cm, 11.69 cm and 11.58 cm, respectively (Table S3).

### 3.4.3. Plant Height

A total of 12 $M_7$ and 137 $M_7$ lines were selected from 17 $M_7$ and 222 $M_7$ mutant lines of NN-1 and Pb-11, respectively. Mean PH of the selected NN-1 mutant lines was 90.24 cm versus 89.26 cm for the wild type (Table 3). Two lines (1621 and 1875) were 95.65 cm and 92.23 cm tall, respectively (Table S2). For Pb-11, the PH mean of the selected mutant lines was 88.37 cm versus 87.65 cm for the wild type (Table 3). Among these, three mutant lines (P-684, P-3144 and P-3163) were 95.65 cm, 95.45 cm and 94.65 cm tall, respectively (Table S3).

### 3.4.4. Tillers per Plant

In total, six $M_7$ mutant lines of NN-1 and 52 $M_7$ mutant lines of Pb-11 were selected. For NN-1, the mean of selected mutant lines for TPP was 5.23 tillers versus 5.0 TPP for wild type (Table 3) or 16.66% greater than the control lines. A total of three lines (284, 320 and 1621) produced six TPP each (Table S2). For Pb-11, the mean of selected lines was 4.68 TPP versus four for the wild type (Table 3). As a result, lines P-684, P-3144, P-3175, P-3176, P-3263, P-3192, P-3231, P-3279, P-2943, P-3146 and P-3215 produced higher number of tillers (5) versus four for the wild type (Table S3).

### 3.5. Correlation among Traits of the Selected Mutant Lines

Mutant lines derived from NN-1 showed a significant positive correlation between DTH and PH, and strong positive correlation with SL and TPP, whereas mutant lines of Pb-11 showed a strong positive correlation of PH with SL and TPP. In both mutant populations, SL showed a highly significant positive correlation with TPP (Figure 3).

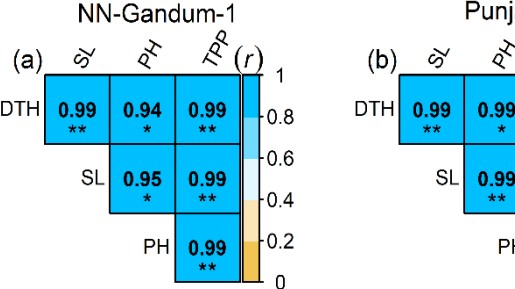

**Figure 3.** Pearson's product-moment correlation (*r*) between selected plants for NN-1 (**a**) and Pb-11 (**b**). DTH = days to heading; PH = plant height; SL = spike length; TPP = tillers per plant; * and ** = significance of the correlation at 0.05 and 0.01 probability (*p*) level, respectively.

### 3.6. Mutagen Effectiveness

Mutagenic efficiency of EMS and γ rays was compared for both the mutant populations. We found that EMS generated 10.47% more resistant/susceptible LR variants compared to γ rays, while γ rays produced an 8.09% greater number of mutants conferring resistance/susceptibility to YR than EMS. For other traits such as PH, SL and DTH, γ rays were 15.55%, 8.70% and 2.32% more effective than EMS, respectively. While in the case of TPP, EMS generated 3.66% more mutants than γ rays. Cumulatively, γ rays produced 34.66% more mutants for all the studied traits (Table 4).

**Table 4.** Comparisons of NN-1 (treated with EMS) and Pb-11 (treated with γ rays) in M7 populations.

| Traits | Variants (%) in $M_7$ Generation | | | | | |
|---|---|---|---|---|---|---|
| | NN-1 (*n* = 3502) | | Pb-11 (*n* = 3453) | | Gain over EMS/γ Rays | |
| | Variants | % | Variants | % | NN-1 (EMS) | Pb-11 (γ Rays) |
| DTH | 590 | 16.85 | 662 | 19.17 | | 2.32 |
| LR | 1172 | 33.47 | 794 | 22.99 | 10.47 | |
| YR | 587 | 16.76 | 858 | 24.85 | | 8.09 |
| SL | 813 | 23.22 | 1102 | 31.91 | | 8.7 |
| PH | 430 | 12.28 | 961 | 27.83 | | 15.55 |
| TPP | 1190 | 33.98 | 1047 | 30.32 | 3.66 | |
| Average | | 22.76 | | 26.18 | | |

DTH = days to heading; LR = leaf rust; YR = yellow rust; SL = spike length; PH = plant height; TPP = tillers per plant; *n* = number of mutant lines.

### 3.7. Heritability

In the present study, a high heritability for NN-Gandum-1 was observed for the characters like plant height (91.66%) and spike length (89.62%) in all three generations (Figure 4). Across all the three generations, the lowest heritability was observed in leaf and yellow rust disease of 21.92% and 27.98%, respectively (Figure 4).

Similarly, high heritability for Punjab-11 was observed for characters such as plant height, e.g., 98.54% in $M_5$ generation, 97.00% in $M_6$ generation and 99.50% in $M_7$ generation, followed by spike length (96.21% in $M_5$, 94.23% in $M_6$ and 92.3% in $M_7$ generations) (Figure 4). Across all the three generations, the highest heritability was observed in plant height (98.35%) and the lowest in leaf rust disease (20.63%) (Figure 4).

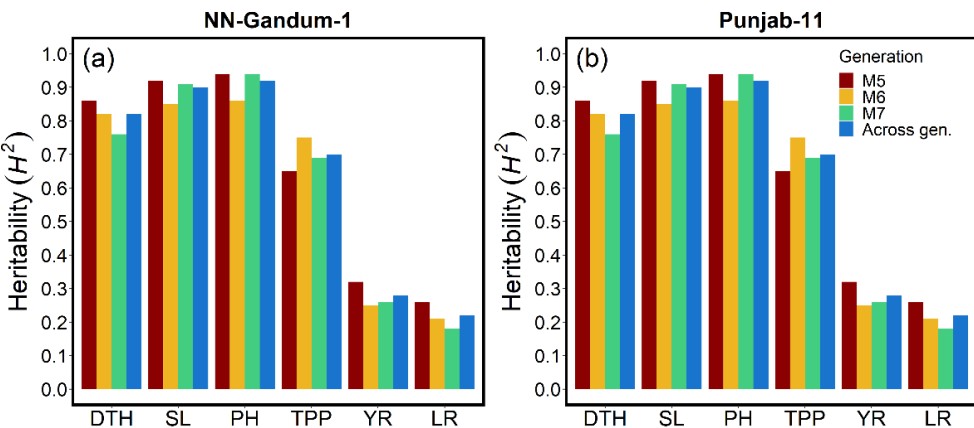

**Figure 4.** Broad sense heritability of the investigated traits in (**a**) spring wheat mutant populations NN Gandum-1, and (**b**) Punjab-11 calculated across three consecutive mutagenized generations. DTH = days to heading; LR = leaf rust; YR = yellow rust; PH = plant height; SL = spike length; TPP = tillers per plant. Color code is given in the legend.

## 4. Discussion

A successful breeding program is largely driven by the genetic diversity found in the germplasm which can be utilized for transferring desirable genes/alleles into the improved cultivars. There are several ways of improving the genetic variability of a given germplasm including collection of new genetic resources from the natural habitats, inducing mutations in germplasm through artificial means, and using modern genomic tools for adding alien genes and editing genes. In the present study, we used a noncontroversial procedure for inducing mutations in wheat genome using chemical as well as physical mutagens. Success of the mutation breeding in creating genotypic as well as phenotypic diversity was demonstrated by developing a first mutant variety of tobacco "Chlorina" [30]. Later on, approximately 3365 crop varieties were released using various mutagens (https://mvd.iaea.org accessed 28 June 2021).

In the present study, 3706 and 3700 $M_1$ plants were generated of both mutagenized populations. It is difficult to estimate the size of a mutant population for launching a successful mutation breeding program. Previously, a varying number of $M_1/M_2$ plants were generated even for the same crop species. For example, $M_2$ populations of *T. aestivum* of the sizes 1532, 1610, 2296 and 3100 plants were developed for identifying desirable mutants [31–34]. In multiple mutagenesis experiments, $M_2$ population sizes ranging from 1632 to 4568 in *O. sativa* were reported [35–38]. Similarly, 4185 and 9575 $M_2$ plants were developed for *S. bicolor* and *H. vulgare*, respectively [39,40].

Here, we assessed >3500 lines of NN-1 and Pb-11 populations in three consecutive generations, viz., $M_5$-$M_7$ in three years. For NN-1, the frequency of phenotypic variation was 22.73%. In multiple investigations, 0.2 to 38.1% phenotypic variants were reported in different wheat mutant populations developed by exposure to EMS [41–44]. In other crop species, mutant populations developed through exposure to chemical mutagens resulted in several mutants. For instance, 33% variants in barley [45], 50% variants in rice [46] and 5% variants in sunflower mutant populations were reported [47]. Chemical mutagens usually induce point mutations, and a few impact the plant phenotype [48].

The other class of mutagens, i.e., physical mutagens, tend to induce larger deletions in genomes, and thus can easily be scored. Here, the proportion of mutants were 26.18% for the mutant population derived from Pb-11. In several investigations, fluctuations in the frequency of mutants were reported. For example, in a wheat mutant population developed through exposing with γ-radiation, a wide range of phenotypic variants for $M_2$ (9.59–11.86%) and its $M_3$ (4–4.87%) populations were reported [48]. Similarly, 8.35–44% phenotypic variations were identified in an $M_3$ *Brachypodium* mutant population developed

through γ-radiation [45]. These variations in the frequency of mutants may be attributed to the type of genotype, dose of a mutagen, and several environmental factors [48,49].

The mutagenized populations (NN-1 and Pb-11) showed extensive diversity in morphological and agronomic traits as compared to their respective parent genotypes (wild type). In $M_5$ versus $M_6$ of the selected mutant lines of NN-1 (in total 17) and Pb-11 (in total 222), days to heading (DTH) showed a significant positive correlation with plant height (PH), spike length (SL) and tillers per plant (TPP), while, in Pb-11-derived mutant lines, PH showed a strong significant ($p < 0.0001$) positive correlation with SL and TPP. In both mutant populations, SL showed a strong significant positive correlation with TPP, while LR and YR were strongly correlated with each other and negatively with the other traits. Similar results were found in $M_5$ versus $M_7$ and $M_6$ versus $M_7$. In both mutant populations, correlation with the replicated $M_7$ generation data was greater for $M_6$ than $M_5$ for all traits. Strong positive correlation coefficients revealed that selection based on PH, SL, TPP, LR, YR and DTH had an equal contribution towards increasing yield [50]. It was also reported that variation in correlation among different generations for all traits was because of segregation of the particular genes, environmental factors between different years and locations [51].

One of the major focuses of our study was to improve resistance to LR and YR in NN-1 and Pb-11, and, therefore, the plants were selected on the basis of their response to LR and YR. These resistant mutants can be used as varieties and or breeding material. Several wheat mutants were produced through mutagenesis in several countries including Pakistan, Ukraine, Russia, Kazakhstan, India, and China [14,52,53]. Previously, similar methods were adopted in cotton to select mutant lines showing improved fiber quality. Significant improvement in various quality traits, including micronaire, upper half mean fiber length, length uniformity index, bundle strength, reflectance, elongation, and yellowness were reported [51] by using various statistical tests, e.g., Z-scores.

We selected lines for one trait that also showed other desirable attributes contributing towards final yield. Improved wheat yield, the ultimate objective of all breeding programs, is heavily dependent upon various factors, including tillering capacity, heading, spike length, number of spikes per unit area, grain size and number. In the present study, 9 of 17, and 14 of 222 mutant lines were selected from the mutant populations of NN-1 and Pb-11 for DTH, respectively. The DTH varied from 88 to 97 days for NN-1 and 89 to 95 days for Pb-11. This trait is highly heritable and variable [50]. Similarly, high variability for DTH (13.23%) was reported in the $M_2$ wheat mutant population [54]. In rice, DTH of the mutant population varied from 72 to 99 days and 76 to 112 days in another study under natural day length conditions [55–57].

Spike length (SL) is one of the most important traits which helps in accommodating a greater number of spikelets per spike. Of the selected mutant lines, in total 12 out of 17 mutant lines of NN-1 (7.25 to 15.65 cm) and 81 out of 222 mutant lines of Pb-11 (7.28 to 14.56 cm) expressed variations in SL. In another study, mean SLs of the selected 40 wheat mutant lines under normal and late planting were 12 and 10 cm when treated with EMS, respectively [54]. Differences in SL were also reported in other studies on wheat [54,58–60]. Similar kinds of variations were reported in rice panicle, barley spike, and sorghum tassel when treated with chemical mutagens [38,39,61].

A total of 12 out of 17 (70.5%) mutant lines derived from NN-1 and 137 out of 222 (21.62%) mutant lines of Pb-11 exhibiting variations for PH were selected. Variation in PH ranged from 75.25 to 102.36 cm for NN-1 and 83.54 to 96.65 cm for Pb-11. In previous studies, a 5–10% reduction in PH was observed [50,54,59,60,62]. Particularly, the highest variability was recorded for PH (69.90%) in a wheat mutant population [54].

The number of TPPs may impact final yield positively or negatively, largely depending upon the prevailing environment and inputs. Under high input environments, the number of productive tillers per unit area of the same genotype will be higher than that of the tillers produced under low input environments. Here, we selected in total, six out of 17 and 52

out of 222 mutant lines exhibiting variations for TPP from NN-1 and Pb-11, respectively. In previous reports, increase in TPP were reported in wheat $M_2$ population [54].

Mutagenic efficiency is the proportion of the desirable mutation frequency in relation to damage associated with mutation [63]. It is dependent upon the type of mutagen, genotype and/or both [64]. In the present study, cumulatively, the γ rays-treated population produced 3.43% more mutants for all the traits than the EMS population. In support of this, γ rays produced higher number of chlorophyll mutants than in an EMS population [65]. In tomato, it was reported that γ rays (50–150 Gy) proved to be a more efficient and effective mutagen when followed by 0.05% to 10% EMS treatment [63]. In soybean, results indicated that three mutant populations i.e., two γ radiated (0.20 and 0.25 kGy) and one EMS (0.1%), showed good seedling emergence; while two of them viz., IR-JS-101 (0.25 KGy) and IR-DS-122 (0.20 KGy) performed better than EMS [66]. In a study on *J. curcas*, seeds treated with 5 Kr dose of γ rays, and 1% EMS showed a similar effect, while a 25 Kr dose of γ rays and 4% EMS treatment showed an inhibitory effect for all the characters studied, except seed germination, when compared to other treatments. The results concluded that treatments of γ rays were greater compared to those of EMS treatments [67]. Some findings, however, are not consistent with the above-mentioned reports. For example, high mutagenesis was reported in sesame (*S. indicum* L.) when treated with EMS, i.e., the average effectiveness of EMS was several times higher than that of γ rays [68]. Additionally, EMS was found to be seven times more effective in chickpea than γ rays [69]. In other crops, it was reported that EMS is better at producing useful mutations than γ rays, e.g., as seen in rice [70], lentil [71], and soybean [72]. Similarly, it was also found that EMS-mediated mutagenesis is more amenable to induce mutations in the wheat genome [73]. Hence it is concluded that genotype, as well as type of mutagen and the interaction of each, contribute towards fluctuation in mutagenic efficacy of the mutagen.

Due to their high heritability and correlation with grain yield, agronomic traits can be used as indirect selection criteria during breeding and cultivar development [32]. Though heritability of physiological traits is relatively low [32,74], their incorporation in breeding programs may be useful for cultivar development [75]. Therefore, to accelerate breeding aimed at improving grain yield genetic gains in wheat, it is important to dissect genomic regions influencing physiological traits and design associated markers for strategic breeding.

## 5. Conclusions

Multiple generations and replicated trials and treatments with different mutagens confirmed the genetic basis of the variations observed in different traits. These mutants were explored for variations in different rust resistant genes. For example, 104,779 SNPs were detected in 11 selected mutant lines in exome regions. Additionally, genetics of the observed phenotypic variations in several mutants were explored at length using TILLING, resequencing and/or by exome capturing [48]. The identified discrete genetic variations (alleles) can be introgressed into the cultivated wheat varieties using conventional breeding approaches, as well as by designing DNA markers. It may be concluded that γ ray mutagenesis can effectively be utilized in the development of desirable economical and quality traits along with some degree of tolerance against biotic stresses in wheat. The extent of mutations in the genome can also be estimated by resequencing and or sequencing the exonic regions of the wheat genome. The newly developed mutants are expected to be useful to provide insights into the genetics of several biochemical mechanisms, as well as identifying genes/alleles involved in controlling important traits in cultivated hexaploid wheat.

**Supplementary Materials:** The following are available online at https://www.mdpi.com/article/10.3390/agriculture11070621/s1, Table S1: Summary of all traits- year wise for both mutant population, Table S2: Z-scores of the selected lines of NN-Gandum-1 mutagenized population, Table S3: Z-scores of the selected lines of Punjab-11 mutagenized population.

**Author Contributions:** Conceptualization, M.-u.-R.; methodology, M.-u.-R. and Q.H.M.; formal analysis, M.H., R.K. and Q.H.M.; data curation, M.H., M.G., M.A.I., A.A. and S.Z.; writing—original draft preparation, M.H., M.-u.-R. and Q.H.M.; writing—review and editing, Q.H.M., R.K., S.Z., M.-u.-R. and M.S.R.; visualization, R.K., M.H. and Q.H.M.; funding acquisition, M.-u.-R. All authors have read and agreed to the published version of the manuscript.

**Funding:** The funds for the development of mutant wheat populations were provided by the International Atomic Energy Commission (IAEA), Vienna, Austria through an umbrella project entitled "Developing Germplasm through TILLING in Crop Plants Using Mutation and Genomic Approaches (PAK/5/047). These activities were continued through the support of another project entitled "Characterization of mutants derived from EMS-derived Gandum-1 for rust and drought tolerance for sustaining wheat yield in Pakistan, funded by Agriculture Linkage Program, Pakistan Agriculture Research Council (ALP-PARC, CS049). We are also grateful to the Higher Education Commission (HEC), Islamabad Pakistan for providing a part of the research work through a project entitled "Identification of disease and insect mutant using SSR and next generation sequencing tools (6103-NRPU/R&D/HEC-2016).

**Institutional Review Board Statement:** Not applicable.

**Informed Consent Statement:** Not applicable.

**Data Availability Statement:** Data is contained within the article and Supplementary Files.

**Acknowledgments:** The funds for the development of mutant wheat populations were provided by the International Atomic Energy Commission (IAEA), Vienna, Austria through an umbrella project entitled "Developing Germplasm through TILLING in Crop Plants Using Mutation and Genomic Approaches (PAK/5/047). These activities were continued through the support of another project entitled "Characterization of mutants derived from EMS-derived Gandum-1 for rust and drought tolerance for sustaining wheat yield in Pakistan, funded by Agriculture Linkage Program, Pakistan Agriculture Research Council (ALP-PARC, CS049). We are also grateful to the Higher Education Commission (HEC), Islamabad Pakistan for providing financial support to a part of the present research work through a project entitled "Identification of disease and insect mutant using SSR and next generation sequencing tools (6103-NRPU/R&D/HEC-2016).

**Conflicts of Interest:** The authors declare no conflict of interest. The funding agencies had no role in the design of the experiments, in the collection, analyses or interpretation of the data, in the writing of the manuscript or in the decision to publish the results. Q.H.M. is presently a member of a company. However, this does not limit the availability or sharing of data and material.

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
