# Peer review of "Prospects of Developing Novel Genetic Resources by Chemical and Physical Mutagenesis to Enlarge the Genetic Window in Bread Wheat Varieties"

_agriculture, doi:10.3390/agriculture11070621_

Round 1

Reviewer 1 Report

This manuscript described the development of mutation populations on two common wheat varieties NN‐Gandum‐1 (NN‐1) and Punjab‐11 (Pb‐11) using EMS and γ‐rays. Significant genetic variation on several traits (plant height, heading date, spike morphology, and rust resistances) were observed in the M5, M6, and M7 generations. The authors also compared the mutant effectness of EMS and γ‐rays. The mutants developed in this study may be served as new breeding lines/materials and also useful for wheat functional gene cloning studies. The methods described in the manuscript can be also applied by other wheat breeders and researchers.

In general, the study is well designed and conducted in multiple years. The results are reliable and similar mutations in different traits could be identified in both varieties and two treatments. My major concern is that male sterilties or partial male sterities are often observed in the M1 plants after EMS and γ‐rays treatment. Possible open pollination could be happen if no controled self-pollination was applied. It is very commom some of the genetic variations detected in the mutation populations are derived from natural hybridization. I am wondering if the authors performed genetic background check of the mutants (sampleling) using molecular marker technologies, such as SSR markers? This is crucial for estimation accurate mutagen effectiveness!   

Author Response

Thanks for the comments. We took all possible control measures to avoid the cross pollination as mentioned in Line# 119 in MATS, i.e. “At maturity, one main spike of each plant was bagged, and harvested separately”. We too did the survey with SSRs for avoiding any natural variants but we are planning different MS for this.

Reviewer 2 Report

This is a description of the production of mutants in wheat which gained useful traits. The manuscript has a number of points that are unclear.

Line 22-23 and several other places in the MS, e.g. line 90 and line 115: the reader may get the impression that two varieties were each treated with two methods of mutagenesis, but only quite far into the MS it becomes clear that that is not the case: NN-1 is treated with EMS, Pb-11 with gamma-rays. This should be clear from the beginning.

Once you realise this, the (line 27) ‘genotype x mutagen interaction’ becomes clear: basically, you don’t know whether it is due to the variety or to the mutagen. This is too simple, though, as the biological activity of the mutagens is quite different, so relative differences between types of mutations are likely to be related to the method used. In fact, that is what you do in paragraph 3.4. However, that is not the case for the total amount of variation per variety: you cannot know whether one method is more efficient than the other as the dose of the gamma-ray treatment was somewhat optimized (line 111), but that of the EMS treatment was not, and even if you would have attempted that for EMS as well, it may not be so easy to find equivalent doses beforehand, and it may not be easy to apply them exactly the same on all plants (think of the variation in the distance of plants to the gamma-ray source).

Line 39 Triticum aestivum: I would call it ‘bread wheat’ here just to be clear to the reader.

Line 70 ’genetic divergence’: I prefer ‘genetic variation’

Line 73 mutagenesis is a good way to increase genetic variation in a breeder’s germplasm, but not ‘the most widely used’ (and I don’t think these two references would support that statement). In wheat one also uses introgression from other types of wheat, from durum wheat and from crop-wild relatives.

Line 85: tolerance to large deletion would be higher in large genomes (without reference): I don’t know whether genome size is relevant, but ploidy is: polyploid crops have a greater tolerance to very high doses of mutagens, probably because they have multiple copies of all genes, including essential ones.

Line 86-87: this seems to be an important statement about what makes a mutagen effective, but I don’t understand the sentence. Please reformulate.

Line 89: not three populations, but three generations of two populations

Line 149: you report lines with new resistance to rusts, which may be very important for breeding disease-resistance bread wheat varieties. However, you only state ‘Data of response to leaf and yellow rusts’ and do not explain how you assessed the resistance, in terms of absence of damage, spots, sporulation, leaf yellowing, etc?! On line 158 you select lines based on ‘their reaction to LR and YR’, which is equally vague. In the absence of a clear description of the phenotyping, the value of the results on resistant mutant lines remains completely unclear.

Line 165. The z score combines a disease score with a value for the other traits, as the explanation of the formula suggests? That is not clear to me. If so, it suggests that the resistance is not absolute but rather quantitative. In Table 3 the z scores are given per trait – are these also compound numbers?!

Line 209 if you have few resistant lines then it is not surprising that you have many susceptible lines for NN-1, so the ‘However’ is not needed. Unless you mean that you have many lines that are MORE susceptible than NN-1 itself. This comes back in lines 230-236. You need to clarify this. In fact, the whole manuscript does not contain a single value on the disease resistance/susceptibility, only ‘high resistance – moderately resistant-moderately susceptible – highly susceptible’. In addition, the heritability is very low. I can imagine that a disease test is more variable that measuring e.g. height, but that only increases the need to give numbers.

Table 2 why are the values for Pb-11 given in 4 decimals and those for NN-1 with 2?!

Line 329-336 see my remark about relative and absolute differences between the two mutagens above. I think line 335-336 is an observation, not a conclusion that one method is more efficient. Do you take all variants that are outside the range for the two varieties as variant, so more resistant lines combined with more susceptible? And the same for the other traits? That may be interesting when comparing the methods for absolute numbers but not relevant with regard to the success of finding useful mutants. Then you should present (another) Table with only the mutants that you find useful.

 In the Results and in the Discussion you talk about 17 mutants in NN-1 versus 222 in PB-11 – now those are significantly different numbers between the two methods and completely different from those in Table 4, but there is no Table with these numbers, one has to pick the numbers from the text in line 209 or in the Discussion. I think these numbers, and the large difference between the methods, should be discussed properly. What could be the mechanism that gamma-rays produce more resistant plants? is it because susceptibility genes are knocked out? What mechanisms could be envisaged about the other traits? Is it known in wheat that gamma-rays are more efficient in producing disease-resistant plants, or vice versa?

Line 380 a comparison of mutation frequency or frequency of interesting mutants with other species is not informative as they may have used a different dosage etc.

Line 387 why can large deletions ‘easily be scored’ when you look at the phenotype?

Line 412 ‘these resistant mutants’ are quite hidden in the manuscript, as I mentioned earlier.

Author Response

Pls find attached :replies to the comments"
